# Pulsed Radiofrequency Electromagnetic Fields as Modulators of Inflammation and Wound Healing in Primary Dermal Fibroblasts of Ulcers

**DOI:** 10.3390/bioengineering11040357

**Published:** 2024-04-05

**Authors:** Erica Costantini, Lisa Aielli, Giulio Gualdi, Manuela Baronio, Paola Monari, Paolo Amerio, Marcella Reale

**Affiliations:** 1Department of Medicine and Aging Sciences, University “G. d’Annunzio”, 66100 Chieti, Italy; giulio.gualdi@unich.it (G.G.); paolo.amerio@unich.it (P.A.); 2Department of Innovative Technologies in Medicine and Dentistry, University “G. d’Annunzio”, 66100 Chieti, Italy; lisa.aielli@unich.it (L.A.); mreale@unich.it (M.R.); 3Pediatrics Clinic and Institute for Molecular Medicine A. Novivelli, Department of Clinical and Expermental Sciences, University of Brescia and ASST-Spedali Civili of Brescia, 25123 Brescia, Italy; manuela.baronio@unibs.it; 4Department of Dermatology, Spedali Civili of Brescia, 25123 Brescia, Italy; paola.monari@libero.it

**Keywords:** pulsed radiofrequency electromagnetic field, wound healing, dermal fibroblasts, inflammation

## Abstract

Venous leg ulcers are one of the most common nonhealing conditions and represent an important clinical problem. The application of pulsed radiofrequency electromagnetic fields (PRF-EMFs), already applied for pain, inflammation, and new tissue formation, can represent a promising approach for venous leg ulcer amelioration. This study aims to evaluate the effect of PRF-EMF exposure on the inflammatory, antioxidant, cell proliferation, and wound healing characteristics of human primary dermal fibroblasts collected from venous leg ulcer patients. The cells’ proliferative and migratory abilities were evaluated by means of a BrdU assay and scratch assay, respectively. The inflammatory response was investigated through TNFα, TGFβ, COX2, IL6, and IL1β gene expression analysis and PGE2 and IL1β production, while the antioxidant activity was tested by measuring GSH, GSSG, tGSH, and GR levels. This study emphasizes the ability of PRF-EMFs to modulate the TGFβ, COX2, IL6, IL1β, and TNFα gene expression in exposed ulcers. Moreover, it confirms the improvement of the proliferative index and wound healing ability presented by PRF-EMFs. In conclusion, exposure to PRF-EMFs can represent a strategy to help tissue repair, regulating mediators involved in the wound healing process.

## 1. Introduction

Chronic wounds have an important impact on global health [1]. Leg venous ulcers (VLU) account for 60 to 80% of leg ulcers, which are described as the most frequent type of chronic skin wound [2]. The alteration of the wound healing process in these ulcers may persist for weeks or years and can become chronic, leading to the establishment of nonhealing wounds and to the development of complications such as cardiovascular diseases, diabetes, and bacterial infection, thus leading to the worsening of the patient’s quality of life [3,4].

Wound healing (WH) goes through several overlapping and consecutive phases, including hemostasis, inflammation, new tissue formation, and tissue remodeling, in a well-coordinated process, with the active involvement of platelets, immune cells (neutrophils and macrophages) and fibroblasts [5]. Fibroblasts display a prominent role in the wound healing process, contributing to the creation of a new extracellular matrix (ECM) and the deposition of collagen structures, as well as removing denatured proteins and matrix-associated materials not needed for the healing, thanks to the production of proteinases to support the migration and activity of immune cells, vascular cells, and organ-specific cells [6]. Furthermore, fibroblasts produce various cytokines and growth factors that can promote or suppress inflammation, depending on the stage of healing and the specific signals from the surrounding cells. Fibroblasts may play a key role in inflammatory signaling pathway regulation, managing the interplay between inflammatory cells, inflammatory cytokines and growth factors in several pathophysiological processes [7,8].

Nonhealing wounds display a reduced cellular proliferation and unbalanced production of inflammatory cytokines, such as interleukin (IL1), (IL6), and tumor necrosis factor (TNF)α, as well as of growth factors, such as transforming growth factor (TGF)β, platelet-derived growth factor (PDGF), epidermal growth factor (EGF), basic fibroblast growth factor (bFGF), and matrix metalloproteinases (MMPs) [9].

Oxidative damage is another feature in nonhealing wounds; this process can prolong microenvironmental homeostasis disruption [10].

All of the mentioned conditions cooperate to determine the pathological nonhealing in VLU and represent the targets of the therapeutical approach.

Currently, the standard therapy for VLU is local wound management, including debridement, dressing techniques, compression therapy [11], and biofilm and bacterial overgrowth control [12]. In non-responsive cases, the standard therapy is associated with advanced treatments. The advanced WH technologies act on tissue, inflammation/infection, moisture, and edge/epithelialization, referred to using the acronym “TIME”. The most used techniques are negative pressure wound therapy, stem-cell therapy, the application of 3D hydrogel dressings, and oxygen therapy, alongside other remedies to better support the repair process [3,13].

Numerous clinical and in vitro studies have shown that electromagnetic therapy (EMT), including electromagnetic fields (EMFs), extremely low-frequency electromagnetic fields (ELF-EMFs) and pulsed radiofrequency radiation (PRF), could be a notable option in the treatment of different medical conditions.

To date, the clinical efficacy of PRF-EMFs has been observed in bone [14], joint, muscle, and soft tissue injuries, leading to a reduction in pain [15,16].

The pulsed signal generated by PRF-EMFs allows heat to dissipate, preventing excessive heat buildup, and exerts biological effects without causing important structural alterations. PRF-EMFs can induce biological changes such as the enhancement of endogenous bioelectrical currents [2,17], with Ca^2+^ efflux changes and the modulation of pathways involved in inflammatory responses [18,19,20].

Although specific intensities and frequencies are applied to help in the treatment of some health conditions, it is difficult to develop standardized treatment protocols due to the high variability of physical parameters and clinical variables, including the frequency and duration of therapy [14,21].

Despite the advanced knowledge and the widespread therapeutic application of these techniques, the complete mechanism of EMT, and above all of PRF-EMFs, is unclear.

Herein, we aimed to evaluate the mechanisms underlying the effect of a commercial medical device (generating a PRF-EMF) on cell proliferation and migration, the expression of tissue repair mediators and the production of antioxidant molecule in primary human dermal fibroblasts (HDFs) collected from patients affected by VLU.

## 2. Materials and Methods

### 2.1. Patients and Tissue Samples

Eleven patients (36.6% female and 63.4% male, mean age = 55 ± 13) were enrolled from the Department of Dermatology, Spedali civili di Brescia, Brescia, Italy. Patients affected by stable VLU, unresponsive to traditional dressings, were selected for the study. The exclusion criteria included the presence of infective, arterial, inflammatory or diabetic diseases. Patients were treated according to the “Nested graft” technique, which involves the acquisition of numerous punch biopsies from the uninvolved skin and seeding in pits made with other punch biopsies at the edge of venous ulcers of the leg of each patient [22,23]. The skin samples derived from the ulcer’s edge (destined to be thrown away) constituted the study sample, while punches from healthy skin constituted the internal sample control. Informed consent was obtained from all patients, in accordance with the 1964 Declaration of Helsinki and its subsequent amendments. This study was approved by the internal local ethics committee and approved and supported by Scientific Committee of Sidemast (Società Italiana Dermatologia e Malattie Sessualmente Trasmesse).

### 2.2. Cell Culture

Immediately after collection, the biopsy samples were placed in trypsin at a ratio of 1:3 with Dulbecco’s phosphate-balanced solution (DPBS) (Merk, St. Louis, MO, USA) for the exclusion of epidermis and adipose tissue residues. Biopsies were cut into fragments of about 2 × 1 mm (length by width), washed in DPBS and placed in 35 mm culture plates in the presence of Dulbecco’s modified Eagle medium (DMEM, Merk, St. Louis, MO, USA) supplemented with 10% fetal bovine serum (FBS) (Merk, St. Louis, MO, USA) and 0.5% penicillin-streptomycin (Merk, St. Louis, MO, USA), and then were incubated at 37 °C in a humid atmosphere with 5% CO_2_. The culture medium was replaced every 3 days. After about 2 weeks, each biopsy section spontaneously released fibroblasts that began to proliferate.

Once the cells started growing, skin fibroblasts were synchronized by being placed in 0.1% serum for 48 h before being trypsinized and plated in the presence of complete medium (DMEM with 10% FBS).

In order to avoid any effect deriving from the native environment, all skin fibroblasts were cultured under the same in vitro conditions for five passages. Ulcer fibroblasts (ulcer-HDFs), established in cultures from biopsies of the edge of chronic VLU, were tested and compared side by side in the same experiment with normal fibroblasts (normal-HDFs) grown from biopsies from normal skin. Conventional phase-contrast light microscopy (Leica DMi1, Wetzlar, Germany, obj. ×10) was used daily to assess the morphological features of normal-HDFs and ulcer-HDFs throughout all growth phases.

### 2.3. Pulsed Radiofrequency Electromagnetic Field Device

Ulcer-HDF cultures were exposed to a PRF-EMF generated by a commercially available medical device provided by Tecnica Scientifica Service (TSS) Medical Srl, Turin, Italy. The device emits a PRF-EMF which induces a small constant electric charge over time for the purposes of its internal functioning; hence, its RF emissions are very low and do not cause interference with nearby electronic devices. The circuit that constitutes the PRF-EMF device is powered by a direct current provided by a CR2032 lithium battery with a nominal voltage of 3 V.

The same circuit converts the delivered square wave, emitted in packets of sinusoids, in PRF. The device’s power is < 3 mW, and it does lead to an increase in local temperature. The PRF-EMF emits non-ionizing radiation at a carrier frequency of 27.1 MHz (37 ns) with a carrier RF modulated through a pulse at 600 Hz (1.66) and a duty cycle of 10%. The duration of a single pulse is 167 µs. The load adapted to the antenna output is identical to the parallel between a 5 ohm resistor and a capacity of 150 pF. The PRF-EMF device also has the following specifications: height, 12 cm; antenna width, 5–6 cm; antenna material, copper wire; action depth, 5–7 cm; max thickness, 1 cm; electromagnetic compatibility level, Group 2 class A.

### 2.4. BrdU Assay

Normal-HDFs and ulcer-HDFs (3 × 10^3^) were cultured in growth media (DMEM supplied with 10% FBS and 0.5% penicillin-streptomycin) in 96-well plates. After 24 h (~70% confluence), cell cycle synchronization was performed by means of overnight serum starvation (serum free culture). Once the cell culture preparation phase was concluded, fresh complete medium was added and the proliferation at different time points (3-6-24 and 48 h) was determined by measuring bromodeoxyuridine (BrdU) incorporated into DNA, following BrdU Roche’s colorimetric protocol (Roche, Mannheim, Germany). BrdU incorporation was measured using the GloMax Multi-Detection System (Promega Corporation, Madison, WI, USA) at an absorbance of 450 nm. For ulcer-HDFs, a second culture plate was set up and exposed to the PRF-EMF for 6 h. Proliferation was assessed with BrdU, starting from the same concentration of cells (3 × 10^3^) and at the same time points. All the experiments were performed in triplicate.

### 2.5. Wound Healing Assay and Image Acquisition

The wound healing assay was performed on normal-HDFs and ulcer-HDFs to test their damage repair capacity. A total of 1.6 × 10^5^ cells/35 mm were plated, and after 48 h at 37 °C cells adhered and spread, obtaining a confluent monolayer. Cell cultures were scratched with a straight line across the center of the well with a p10 sterile pipette tip. After scratching, one wash with DPBS was performed to remove debris and fresh medium was added. Culture plates were then placed in the cell culture incubator for 24 h. The ulcer-HDFs were exposed, in a different set of culture plates, to a 6 h period of PRF-EMF.

Plates were observed using a phase-contrast microscope (Leica DMi1, Wetzlar, Germany,), and the edges of the induced wound area (scratch) were documented, acquiring pictures with a digital camera (Leica DMi1, Wetzlar, Germany,) at 0 h, 6 h, and 24 h to evaluate the fibroblasts migration. The images were processed using the NIH ImageJ software version 1.54 h [24] to calculate the wound area dimensions. The data were obtained from triplicate experiments.

### 2.6. Gene Expression Profiling

Total RNA was isolated using QIAzol reagent (Qiagen, Hilden, Germany) and reverse transcribed with the QuantiTec Reverse Transcription kit (Qiagen, Hilden, Germany) according to the manufacturer’s instructions. Real-time qPCR was performed using GoTaq^®^ qPCR Master Mix (Promega, Milan, Italy) and a Bio-Rad Real-Time PCR instrument (CFX Real-Time PCR Bio-Rad, Hercules, CA, USA) with the following cycling conditions: 95 °C for 10 min, followed by 40 cycles of denaturation at 95 °C for 10 s, annealing at 60 °C for 10 s, and extension at 72 °C for 20 s. The primer sequences used for qPCR are provided in Table 1. qPCR results were analyzed using Bio-Rad system software (CFX Manager). The 2^−∆∆Ct^ method was used to detect the relative expression of TNFα, TGFβ, cyclooxygenase (COX)2, IL6 and IL1β, using RPS18 to normalize the gene expression levels. Relative quantification cycle (Ct) values were reported as fold changes in expression. Experiments were performed in triplicate and the data were averaged.

### 2.7. ELISA Assay

The concentration of IL1β and prostaglandin (PG)E2 was assessed in the supernatant of normal-HDFs, ulcer-HDFs, and PRF-EMF-exposed ulcer-HDFs using the Enzyme-Linked ImmunoSorbent Assay (ELISA). Specifically, after each experimental protocol, the cell culture supernatant was collected and stored at −80 °C for subsequent evaluation. Before assessing the IL1β and PGE2 levels, samples were centrifugated at 10,000× *g* for 5 min to eliminate cell debris and they were then plated following the manufacturer’s instructions. Relative absorbance was measured at 450 nm using the GloMax Multi-Detection System (Promega Corporation, Madison, WI, USA). Cytokine concentration was calculated using a standard reference curve. The intra- and inter-assay reproducibility was >90%. The specificity and the sensitivity of the cytokine were defined according to the manufacturer’s guidelines.

### 2.8. Antioxidant Mediators Quantification

The amount of total GSH (tGSH), oxidized GSH (GSSG), and free GSH (GSH) in the cell culture supernatant was quantified using a colorimetric detection kit for Glutathione (Arbor Assays, Ann Arbor, MI, USA), while for glutathione reductase (GR), a fluorescent activity kit was used (Arbor Assays, Ann Arbor, MI, USA). GSH was calculated by subtracting GSSG from the total fraction, where the oxidized data were obtained using 2-vinylpyridine to block the free fraction in the samples. Experiments for each different condition (normal-HDFs, ulcer-HDFs, and exposed ulcer-HDFs) were conducted in duplicate.

### 2.9. Statistics

GraphPad Prism (v.6.0; GraphPad Software, La Jolla, CA, USA) was used for statistical analysis of the data. All results were expressed as mean ± SD. For repeated measures, one-way ANOVA was performed to compare differences between groups. The differences between the normal-HDFs, ulcer-HDFs, and exposed ulcer-HDFs were measured by means of Tukey post hoc comparison or by Student’s t-test for unpaired data. Significant differences were established at *p* < 0.05.

## 3. Results

### 3.1. Cell Morphology

Using light microscopy in the routine monitoring of cell cultures, differences in morphology and growth rate between normal-HDFs and ulcer-HDFs were highlighted. In normal-HDFs, the increased number of cells was readily apparent when cultures were viewed under a light microscope. Cells appear with a normal morphology, being compact with a spindle shape and well-defined nuclear morphologic features. Meanwhile, the ulcer-HDFs appear larger with a polygonal shape, including some lipid droplets and granular cytoplasmic structures, with nonuniform nuclear morphologic features such as segmented nucleoli. Furthermore, starting from the same density of plated cells, and observing the cultures after 6 and 24 h, cells are differently distributed in the well. Indeed, a reduction in the growth rate of ulcer-HDFs after only 6 h was observed, in accordance with the literature data [25,26]. After 24 h, differences in concentration, density, size, arrangement and orientation of ulcer-HDFs were still detectable with respect to normal-HDFs (Figure 1).

### 3.2. BrdU Assay

The ability of normal-HDFs and ulcer-HDFs in cell proliferation was assessed by means of the BrdU uptake assay, which is a marker for cell proliferation due to it being rapidly taken up and accumulated by dividing cells since it is not metabolizable. Our results show that, starting from the same cell concentration (3 × 10^3^), normal-HDFs have an increased proliferation rate compared to ulcer-HDFs. The proliferation rate is significantly enhanced for all measured time points (3, 6, 24 and 48 h), with a rapid increase in BrdU accumulation in normal-HDFs immediately after 3 h and with a steady increase at 6 and 24 h.

Otherwise, in ulcer-HDFs, a slow incorporation of BrdU is observed, with an increase after 6 h (ratio = 1.7 vs. 0 h) and an additional increase at 24 h (ratio = 1.9 vs. 0 h) in comparison with the basal levels. For both normal- and ulcer-HDFs after 48 h, there is a slight reduction in cell proliferation (Figure 2). These data underline that cells isolated from the ulcer area show a slower replicative capacity and a longer time to become confluent when compared with the healthy skin fibroblasts.

Recently, several studies [5,27,28,29] have suggested that the application of a PRF-EMF modulates fibroblasts’ capability to regulate tissue homeostasis. To investigate the ability of a PRF-EMF to reduce the proliferative gap between ulcer-HDFs and normal-HDFs, the ulcer-HDFs of each patient were seeded at a concentration of 3 × 10^3^ in 96-well plates and exposed for 6 h to the PRF-EMF. After 3, 6, 24 and 48 h, BrdU uptake and the proliferation rate were determined. A growth curve with the ratio between the different time points and the 0 h levels of BrdU for each condition is reported in Figure 3. Our results show that ulcer-HDFs’ proliferation rate is significantly lower than that of normal-HDFs, while the exposure of ulcer-HDFs to a PRF-EMF determines an early improvement in proliferation at 3 h compared to the normal-HDFs. This improvement is maintained for the other time points (Figure 3).

### 3.3. Scratch Wound Assay

To evaluate cell migration and the regeneration of the cell monolayer, the most used in vitro model is the mechanical damage model (“scratch wound assay”). The ability of normal-HDFs, ulcer-HDFs and exposed ulcer-HDFs to migrate into the damaged area was evaluated. Immediately (0 h), 6 h, and 24 h after the scratch, pictures were acquired and processed with the ImageJ software to calculate the size of the damaged area. The size of the initial scratch was calculated and assumed as 100% of the cell-free area for each sample and condition. After 6 h, in normal-HDFs, the cell-free area was 68.6%, and after 24 h, it was only 8.2%. In accordance with the decreased proliferation, the reduction in the cell-free area in scratched ulcer-HDFs is lower than that in normal-HDFs, with the cell-free area being 75.1% after 6 h and 10.4% after 24 h, compared with 100% after 0 h (Figure 4).

To evaluate if PRF-EMF exposure can affect ulcer-HDFs’ migration for wound closure, cells were scratched and exposed to the PRF-EMF for 6 h. Observing the exposed ulcer-HDFs after 6 h, the cell-free area was 63.5%, compared to 100% at 0 h, with significant differences compared to unexposed ulcer-HDFs (cell-free area of 75.1%; *p* < 0.001) and normal-HDFs (cell-free area of 68.6%; *p* < 0.05) at the same time. Furthermore, 24 h after the scratch in all of the evaluated HDF samples and conditions, a considerable improvement in the regenerative capacity was observed, with a significant reduction in cell-free area in PRF-EMF-exposed ulcer-HDFs with respect to both the normal- (*p* < 0.05) and ulcer-HDFs (*p* < 0.001) (Figure 4b).

### 3.4. Gene Expression

Considering the broad role of inflammatory cytokines in the regulation of the WH process, we evaluated the expression of COX2 and pro-inflammatory cytokines, namely IL1β, IL6, TGFβ, and TNFα, that are important for cell proliferation and the synthesis of the ECM, both in normal-HDFs and ulcer-HDFs. At the end of incubation (24 h), the gene expression of mediators in ulcer-HDFs in comparison with normal-HDFs was significantly higher, in accordance with the persistence of the inflammatory phase in chronic ulcers (Figure 5). Thus, in this study, we evaluated the effect of the exposure to a PRF-EMF on ulcer-HDFs, observing a significant increase in the expression levels of IL1β, IL6, COX2, and TGFβ with respect to unexposed normal-HDFs and ulcer-HDFs.

### 3.5. PGE2 and IL1β Levels

The levels of production of PGE2 and IL1β were evaluated in scratched normal-HDF, ulcer-HDF, and exposed ulcer-HDF supernatants to underline the differences between the cell lines. PGE2, which constitutes the major PGE in human skin [30,31], and IL1β, which is a master cytokine for cell recruitment and activation [32], can be produced by many cell types, such as epithelial cells, fibroblasts, and keratinocytes, as well as inflammatory cells. The production of PGE2 and IL1β increases significantly in the presence of damage and influences cell growth and differentiation processes. Indeed, in our data, we observed increased levels of PGE2 in ulcer-HDFs with respect to normal-HDFs, although this was not significant. When ulcer-HDFs are exposed to a PRF-EMF, in accordance with the results of Cheng et al. [33], we observed a higher increase level of PGE2 with respect to normal-HDFs.

The levels of IL1β show the same trend, with a slight increase in ulcer-HDFs and a more significant increase in exposed ulcer-HDFs with respect to normal-HDFs, in accordance with the increased cell proliferation and early scratch healing progression (Figure 6).

### 3.6. Antioxidant Activity

Oxidative stress, due to an imbalance in the pro-oxidant–antioxidant homeostasis, plays an important role in the nonhealing of wounds. When a higher load of reactive oxygen species (ROS), caused by the abnormal generation of or deficiencies in the antioxidant defenses, persists over a long time, continuous damage and chronic nonhealing wounds are detected. We focused our study on the evaluation of antioxidant mediators, measuring tGSH, GSSG, and GSH, as well as the activity of GR, an enzyme responsible for catalyzing the reduction of GSSG to GSH. Our results show that in comparison with normal-HDFs, in ulcer-HDFs, there are no significant differences in GR activity (0.18 mU/mL in normal-HDFs and 0.19 mU/mL in ulcer-HDFs) or in GSSG levels (0.22 µM and 0.25 µM in normal- and ulcer-HDFs, respectively), while tGSH and GSH levels are significantly reduced (*p* < 0.001), in accordance with the impaired fibroblast proliferation and migration driven by the production of ROS, the lack of antioxidant defenses, and the excessive oxidative stress.

After the PRF-EMF exposure of ulcer-HDFs, the levels of the antioxidant molecules are comparable to those in ulcer-HDFs (Figure 7).

Therefore, since the ratio between GSH and GSSG can represent an important indicator of cell health [34], we calculated the ratio and pointed out that in both ulcer-HDFs and exposed ulcer-HDFs, there is a significant reduction compared to normal-HDFs. Thus, PRF-EMF exposure is unable to modify the antioxidant system.

## 4. Discussion

A complex crosstalk and a different cellular response are involved in the WH process, resulting in the overlap of dynamic phases (hemostasis, inflammation response, new tissue formation, and tissue remodeling). The alteration or deregulation of one or more of these phases may lead to chronic ulcers. Fibroblasts actively participate in WH and orchestrate all of the phases of tissue repair/regeneration process through interactions with other cell populations involved in the process [7]. In this study, we first assessed the differences between HDFs isolated from normal and ulcerative areas of patients affected by VLU, the most frequent form of chronic skin ulcers, and following this we evaluated the effect of a PRF-EMF on ulcer-HDFs’ morphology, proliferation and gene expression, as well as WH modulation.

Significant differences were observed in the morphology and proliferation rate of ulcer-HDFs compared to normal-HDFs, with alterations in shape and a reduced growth time. These characteristics resemble those observed in senescent fibroblasts and confirm the findings of Wall et al., who demonstrated that fibroblasts from chronic nonhealing wounds display abnormal phenotypes, including decreased proliferation, early senescence, and altered patterns of cytokine release [35].

In the last few years, an increasing number of reports have evaluated the effects of ELF-EMFs on keratinocytes and immune cells involved in skin repair. ELF-EMFs act on the WH process though the modulation of inflammation, protease matrix rearrangement, neo-angiogenesis, senescence, stem-cell proliferation, and epithelialization. The exposure times, waveforms, frequencies, and amplitudes used in the different literature reports are very varied and the results obtained are often in contrast, highlighting that the biological effects of ELF-EMFs may vary with the EMF’s physical characteristics and based on the type of target cell [28].

To evaluate the cell migration and regenerative capacity, we applied the widely used in vitro “scratch wound assay”, inducing mechanical damage to confluent cell layers.

PRF-EMF exposure for 6 h led to a significant improvement in the proliferation ability of ulcer-HDFs subjected to the scratch wound assay, which promptly migrated to the wounded area and displayed accelerated wound closure.

The cell-free area caused by the scratch was covered at a proportion of 31.4% by normal-HDFs 6 h after the scratch, while ulcer-HDFs were capable of covering only 24.9% of the wound area. These differences were also maintained at 24 h, when ulcer-HDFs presented an 89% reduction in the cell-free area compared to the 92% reduction noted for the normal-HDFs, supporting the hypothesis regarding the altered proliferation/migration capabilities of ulcer-HDFs.

The pattern displayed by ulcer-HDFs in reaching a confluence layer was dissimilar from that of normal-HDFs. Ulcer-derived HDFs individually adhered to the dish and then randomly migrated, occasionally coming into contact with other cells. We hypothesize that the shape alteration of ulcer-HDFs weakens the cell–cell interaction and may be responsible for their reduced ability to cover the cell-free area.

Interestingly, after exposure to the PRF-EMF, the proliferative index of ulcer-HDFs increased significantly and better cell alignment and movement towards neighboring cells were evident, resulting in optimal wound closure.

The exposure to PRF-EMF prompts an earlier reduction in the scratch-induced cell-free area displayed by exposed ulcer-HDFs (11.6% coverage) compared to the unexposed ulcer-HDFs and even to normal-HDFs (5.1% coverage) (*p* < 0.05). This trend persisted after 24 h, showing that in PRF-EMF-exposed ulcer-derived HDFs, there is an increase in the repair ability.

In agreement with the literature [36,37], a more intense expression of COX2, IL1β, IL6, TGFβ, and TNFα was observed in ulcer-HDFs than in normal-HDFs.

These cytokines are involved not only in the inflammation phase but also in the epithelialization phase, promoting cell proliferation and migration, fibroblast differentiation, and the mobilization of resident stem/progenitor cells [38]. Our study demonstrated an increase in the levels of these cytokines in ulcer-HDFs after PRF-EMF exposure, which may explain the results regarding the increased migration in the scratch wound assay.

We suggest that in inefficient WH, such as in chronic wounds, the exposure to a PRF-EMF may help to restore the well-orchestrated interaction between cells and mediators, driving the progression of overlapping phases of inflammation, proliferation, and tissue remodeling.

Furthermore, we hypothesize that the increase in TGFβ gene expression after PRF-EMF exposure can be responsible for fibroblast and mesenchymal cell activation, as well as the recruitment and activation of neutrophils and macrophages.

This could be important since it is known from the literature that in the early phase of WH perturbation, neutrophil recruitment may induce the alteration of monocyte infiltration timing with decreased IL1β secretion, which in turn reduces keratinocyte migration and proliferation.

Changes in the macrophage phenotype during the healing process help in the transition from a pro-inflammatory to a pro-resolution state, promoting keratinocyte, fibroblast, and epithelial cell proliferation with the secretion of cytokines and growth factors [39].

Interestingly, our finding that that levels of IL1β were significantly higher in the supernatant of exposed ulcer-HDFs with respect to unexposed ulcer-HDFs and normal-HDFs may explain the optimization of the wound closure assay, since evidence [38,40,41] has shown that IL1β levels correlate with active immune cell infiltration following the exacerbation of inflammation, leading to the rebalancing of pro-inflammatory cytokines and aiding in the transition from the inflammatory to the proliferative phase in skin wounds.

Moreover, our study revealed that the overproduction of IL1β correlates with the higher release of PGE2 in PRF-EMF-exposed cells, prompting an increase in cell proliferation [42] and TNFα gene expression inhibition [43,44].

In both unexposed and exposed scratched ulcer-HDFs, we observed decreased activity of antioxidants with respect to scratched normal-HDFs. These data agree with the role of unbalanced oxidant/antioxidant homeostasis, a reduction in GSH levels and alterations in the overall redox status in the worsening of the microenvironment in chronic wounds [45,46]. In our study, in exposed ulcer-HDFs, probably due to the system parameters of the PRF-EMF device, such as the frequency, pulse or intensity, significant modulation of antioxidant activity was observed [47,48].

## 5. Conclusions

The results of this study show that a PRF-EMF may affect ulcer-HDFs’ cell proliferation and modulate the expression and production of cytokines, leading to an improvement in WH. Our results indicate that a PRF-EMF enhances ulcer-HDF activation, helping the WH by activating the robust migration of fibroblasts and by further stimulating the inflammatory response. The recruitment of other cells is necessary to continue the healing process, pushing forward all repair phases and stimulating and coordinating the essential functions of wound repair.

We acknowledge that the transition from two-dimensional (2D) monocultures of dominant cell types such as keratinocytes and fibroblasts to co-culture systems and to more complex three-dimensional (3D) tissue models is needed to improve the transferability of our results.

The exploration of the mechanics and effects PRF-EMF exposure might help in the search for promising approaches for chronic WH treatment. The next goal would be to evaluate the effect of PRF-EMFs, alone and in addition to other standard therapies, in order to investigate additional effects and hypothesize the application of a PRF-EMF as a supportive therapy.

## Figures and Tables

**Figure 1 bioengineering-11-00357-f001:**
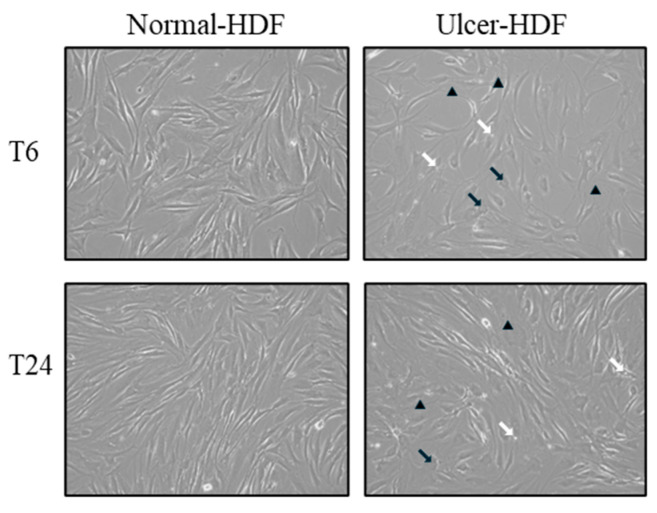
Light microscopy displaying the cellular morphology and confluency of normal-HDFs and ulcer-HDFs. Micrographs are representative images from independent experiments performed for each sample and in duplicate. Images of normal-HDFs and ulcer-HDFs were captured after 6 h and after 24 h of cell incubation. Total magnification = 10×. Lipid droplets are indicated by white arrows, granular cytoplasmic structures are indicated by black arrows, and segmented nucleoli are indicated by black arrowheads.

**Figure 2 bioengineering-11-00357-f002:**
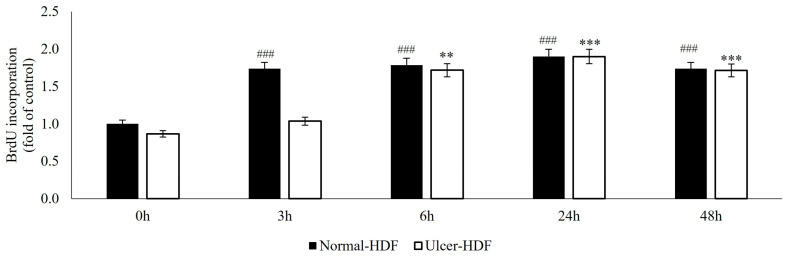
BrdU uptake of normal- and ulcer-HDFs evaluated by the BrdU assay. All experiments were performed in triplicate and the results are presented as fold of the control (normal-HDFs). Statistical significances: ### *p* > 0.001 for time comparison in the normal-HDF group vs. normal-HDF at 0 h; ** *p* > 0.01 and *** *p* > 0.001 for comparison in the ulcer-HDF group vs. ulcer-HDF at 0 h.

**Figure 3 bioengineering-11-00357-f003:**
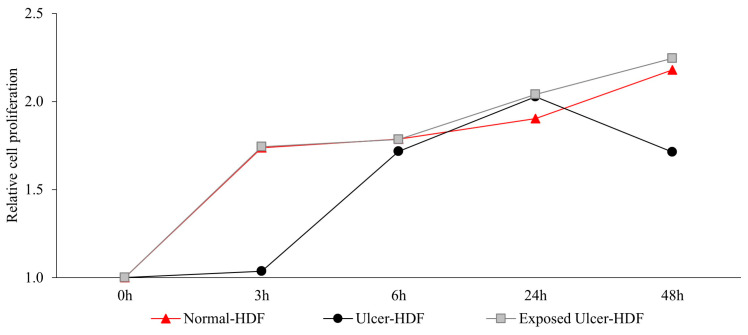
Relative cell proliferation ability in ulcer-HDFs after 6 h of PRF-EMF exposure compared to normal- and ulcer-HDFs without exposure, measured using the BrdU proliferation assay. Relative cell proliferation was calculated as the ratio between each time point and 0 h. All experiments were performed at least three times.

**Figure 4 bioengineering-11-00357-f004:**
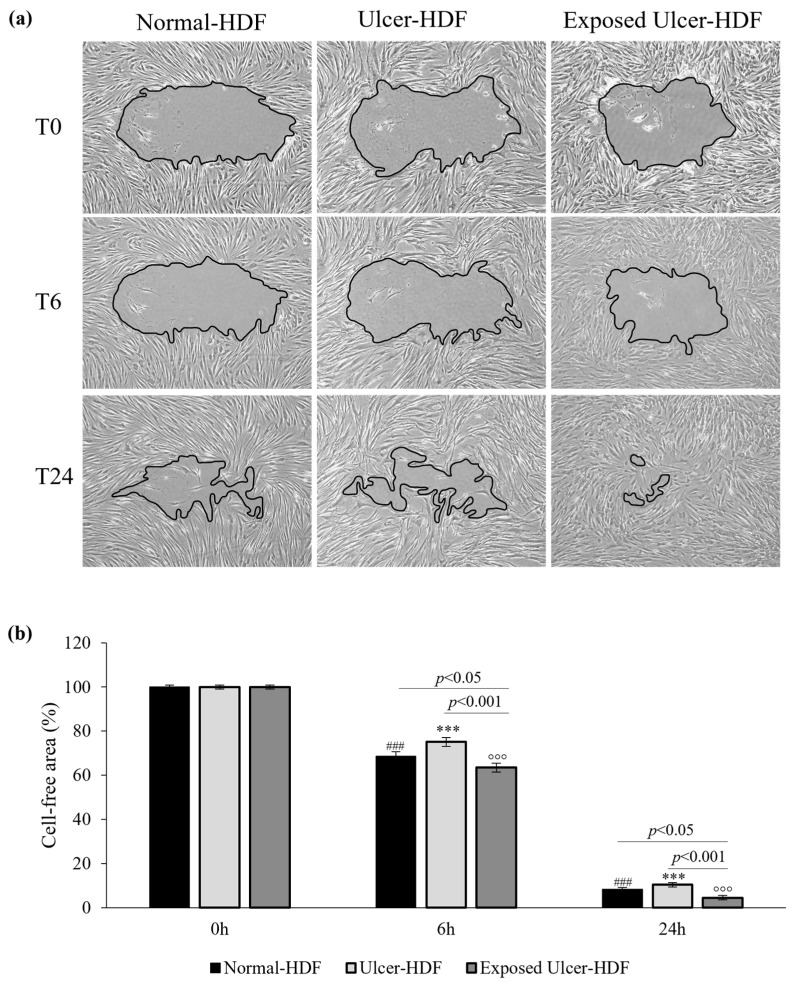
(**a**) Light microscopic images of normal-HDF, ulcer-HDF, and exposed ulcer-HDF scratched cells. Images with 10× magnification were captured at 0 h, immediately after the wound creation, at 6 h post wound, and at 24 h post wound. A Leica DMi1 microscope with a digital camera was used to capture images, and the cell-free area was measured using NIH ImageJ software version 1.54 h. (**b**) Graph of the percentage of cell-free area of the scratched HDFs. One-way ANOVA statistical significance: ### *p* < 0.001 for the comparison with normal-HDFs at 0 h; *** *p* < 0.001 for the comparison with ulcer-HDFs at 0 h; °°° *p* < 0.001 for the comparison with exposed ulcer-HDFs at 0 h. Differences between different samples were considered significant at *p* < 0.05.

**Figure 5 bioengineering-11-00357-f005:**
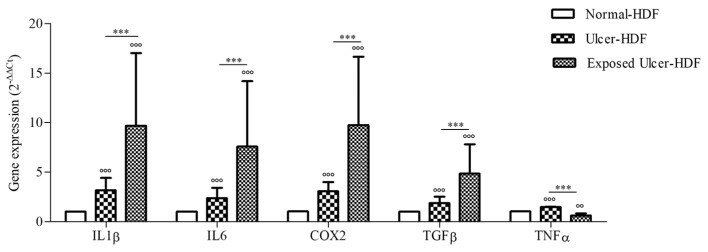
Gene expression of IL1β, IL6, COX2, TGFβ, and TNFα in ulcer-HDFs and exposed ulcer-HDFs compared to normal-HDFs, assumed as 1. Changes in gene expression were determined by means of qPCR and evaluated via the 2^−ΔΔCt^ method. Data are reported as the mean and 95% CI. °°° *p* < 0.001 and °° *p* < 0.01 for the comparison with normal-HDFs; *** *p* < 0.001 in PRF-EMF-exposed ulcer-HDFs compared with ulcer-HDFs.

**Figure 6 bioengineering-11-00357-f006:**
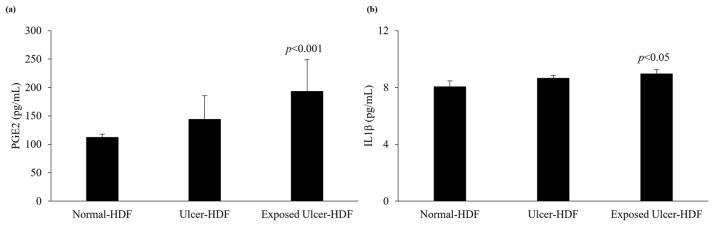
Analysis of levels of (**a**) PGE2 and (**b**) IL1β in normal-HDF, ulcer-HDF, and exposed ulcer-HDF supernatants. Values represent the mean ± SD of three independent experiments. Statistical significance for *p* < 0.05 with respect to normal-HDFs.

**Figure 7 bioengineering-11-00357-f007:**
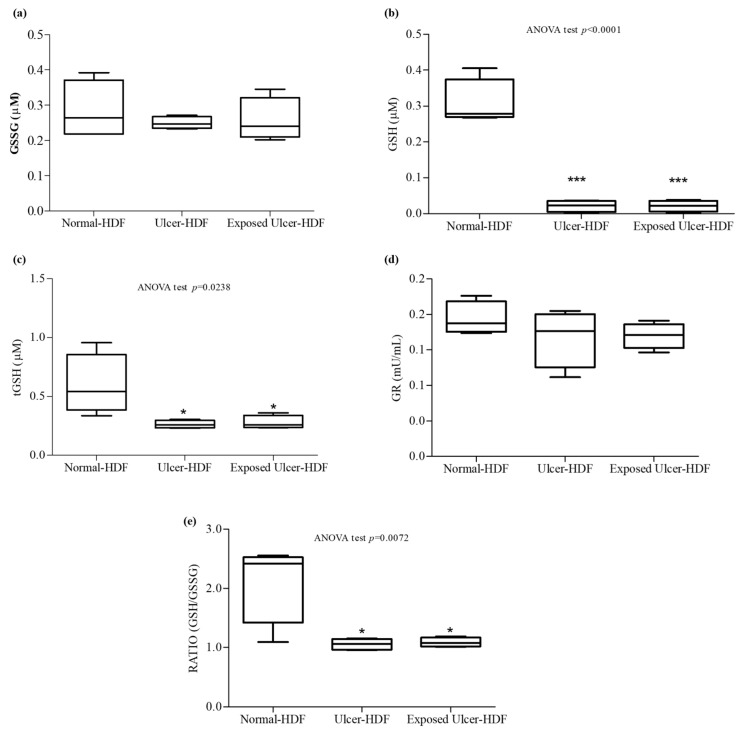
Antioxidant activity. (**a**) Oxidized glutathione (GSSG); (**b**) free glutathione (GSH); (**c**) total glutathione (tGSH); (**d**) glutathione reductase (GR); (**e**) ratio of GSH/GSSG concentrations in the supernatant of wounded normal-HDFs, ulcer-HDFs, and exposed ulcer-HDFs. Whisker plot represents the distribution of numeric data values with the minimum and maximum. Significant differences were detected at * *p* < 0.05 and *** *p* < 0.001 with respect to normal-HDFs.

**Table 1 bioengineering-11-00357-t001:** Primer pair sequences used in the study.

Gene	Forward Primer Sequence (5′–3′)	Reverse Primer Sequence (5′–3′)	Amplicon Leght
TNFα	CCTTCCTGATCGTGGCAG	GCTTGAGGGTTTGCTACAAC	184 bp
TGFβ	AACAATTCCTGGCGATACCTC	GTAGTGAACCCGTTGATGTCC	197 bp
COX2	GACAGTCCACCAACTTACAATG	GGCAATCATCAGGCACAGG	105 bp
IL6	GTACATCCTCGACGGCATC	ACCTCAAACTCCAAAAGACCAG	198 bp
IL1β	TGAGGATGACTTGTTCTTTGAAG	GTGGTGGTCGGAGATTCG	115 bp
RPS18	CTTTGCCATCACTGCCATTAAG	TCCATCCTTTACATCCTTCTGTC	199 bp

## Data Availability

Data are contained within the article.

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
