# Peer review of "Pulsed Radiofrequency Electromagnetic Fields as Modulators of Inflammation and Wound Healing in Primary Dermal Fibroblasts of Ulcers"

_bioengineering, 2024, doi:10.3390/bioengineering11040357_

Round 1

Reviewer 1 Report

Comments and Suggestions for Authors

Comments for Manuscript ID: bioengineering-2891636

The ethical approval paperwork from the hospital setting is missing. The hospital where the tissues were collected remains unspecified.

Venous chronic ulcers usually appear with a ruptured wound and discharge. Did the individuals in the study who provided tissue samples has skin ruptures or fluid discharge?

Figure 3 shows that the cellular proliferation potential of ulcerated HDF is comparable to that of normal HDF. The proliferation rate of ulcer HDF should be very low, as indicated in a previous study (https://doi.org/10.1016/S0741-5214(98)70064-3). Kindly provide a rationale for your findings.

Reviewer 2 Report

Comments and Suggestions for Authors

TITLE

            The title is not well written. There are interesting findings in the study and it should be rewritten.

ABSTRACT

            English must be revised. In line 21, replace ‘wound healing’ response for cell migration, once you did not performed in vivo studies and analyzed wound healing parameters. The assays is called wound healing, just that.

            Line 26 – ‘...TGFβ, COX2, IL6, IL1β and TNFα’ expression, it is important to highlight that; you did not evaluated protein here.

INTRODUCTION

            At first, English must be revised in the entire Introduction section. Phrases are too long and the translation is too poor, with some grammar errors. Regarding the Introduction content, it is well structured with a good line of thought. I will make some suggestions according to the paragraphs:

From Line 36 to 42: I suggest exploring more the physiopathology of the VLU, highlighting their importance and particularities compared to other chronic wounds. It will enrich the justification of the study.

From Line 43 to 51: Replace ‘wound closure’ for ‘wound healing’, it is more accurate. Closure is one of the healing steps. This statement ‘Fibroblasts are prominent cells in the WH, compared to other epithelial and 46 endothelial cell, not only for their ability to break down fibrin clots and form new extracellular matrix (ECM) with collagen structures’ is not suitable. I understood that you want to highlight the role of the fibroblasts, but the other cells play equal important roles on wound healing. Without endothelial cells, the stroma is not vascularized and become hypoxic and epithelial are responsible to form the protection barrier of the skin. There no degrees of importance here. Rewrite! Moreover, ECM formation is not due to fibrin clots breaking, but a great number of ECM molecules are synthetized by fibroblasts, take care for no to be being inaccurate on this matter.

From Line 52 to 59: It is very difficult to read, rewrite this whole paragraph. You mentioned several molecular mechanisms related to cytokines, growth factors and cell proliferation, but forgot to highlight ECM remodeling issues, fibrosis, cell migration impairment and alterations on cytoskeleton. If you decided to explore these mechanisms, you have to provide more details and the ones related to fibroblasts themselves.

From Line 60 to 66:  It is very difficult to read, rewrite this whole paragraph, again. If the whole basis of your study is the modulation of fibroblasts and, thus, ECM remodeling as well, I would like more information about ECM-based treatments for wound healing and if there is any studies for VLU.

From Line 67 to 72: If they are clinical trials, it is obvious they are in vivo. It is redundant here. Again, the English is sufferable. It is difficult to understand. Native speaker review is demanding here. The effects of EMF on cell and tissue behavior are very vague here. I would like more information to understand the application of such technologies.

From Line 73 to 75: Poorly written. It needs more information about the difficult to standardize the treatment protocols.

From Line 76 to 82: Again, English is a major problem. You quote several applications of PFR-EMT in several pathologies, but provide only one reference. Where are the other references? Take care with that.

From Line 83 to 86: Here is your justification for the study. Rewrite in a manner that provides some piece of information about your study outcomes, not summarizing the methodology.

MATERIALS AND METHODS

Regarding the section ‘2.1. Patients and tissue samples’, which is the location where the study were conducted, and where is the protocol number of the study approved by the Ethical Commission of Studies with Humans. Without these data, your study cannot proceed to publication if accepted. I recommend immediate addition of such information.

            Regarding the section ‘2.2. Cell Culture’, how this fibroblasts were characterized from the biopsies? The skin is very heterogeneous and to secure a pure culture, once cell contamination can compromise your data. Explain this better in this section.

            Is this phrase correct? ‘The duration of the 125 single pulse is 167μm.’ Duration should be in seconds or minutes, is that correct? Please, elucidate this.

RESULTS

            Regarding Figure 1, you describe several morphological alterations on ulcer-HDF as ‘some vacuole and detritus, with not 205 uniform nuclear morphologic features’; it is not very clear your statement with the figure. I recommend highlight on the figure with some arrows, circles or symbols and add to the caption. This will make your findings unquestionable.

            A concern was raised regarding the cell scratch assays. It is visible that the ‘scratch’ of the Exposed Ulcer-HDF is smaller than the other groups, which makes the comparison biased. Could you provide the raw quantification data of the cell-free areas to attest a correct comparison, because if the initial scratch were smaller, the cell repopulation obviously would be higher, right? I would like to see this data properly; maybe you chose a wrong representative image to exemplify the experimental condition.

DISCUSSION

            English review urgently. Besides that, remove or rearrange lines 379-380, you are repeating the objectives if the study without connection with the rest of the ‘Discussion’ text.

            Again, I raise some concerns about the cell scratch experiment. Please, provide the raw information and evaluate the conclusions from the experiment, just a quick reminding.

            From line 414 to 420, it is completely unintelligible; rewrite that because it is confusing.

            How do u explain a higher ILβ1 gene expression in exposed ulcer-HDF, but no difference in the ILβ1 ELISA results? Elaborate.  

            The last paragraph, this explanation of antioxidants alterations are not very clear. I would like more literature of other studies that used PFR-EMF and performed an antioxidant evaluation and correlate with your results. Antioxidants are of major importance in wound healing and, according to your results; the exposure did not revealed any outstanding results.

CONCLUSION

            Your conclusion is extensive, but it is appropriated for the study. You pointed an important limitation of your study, which is the 2D system used. 3D culture systems can alter completely cellular response to any treatment and the cell-ECM as well. Consider scaffolds, ECM hydrogels to improve your culture system complexity. Be aware that this results are preliminary in vitro studies, so take care to be bold on you conclusion regarding the possible clinical outcomes. These alternative therapies must be observe carefully, once their variables may become the treatment outcomes unpredictable and inconsistent. I recommend in the future in vivo animal trials.

Comments on the Quality of English Language

English must undergo an extensive review. Long phases and erroneous translation difficult the reading. 

Round 2

Reviewer 1 Report

Comments and Suggestions for Authors

Comments for Manuscript ID: bioengineering-2891636

The manuscript titled Primary dermal fibroblasts of ulcer upon wounding and exposure to pulsed radiofrequency electromagnetic field” can be accepted in the present form.

Author Response

The authors thank for the approval. 

Reviewer 2 Report

Comments and Suggestions for Authors

The authors agreeded with my requests, performed signficant changes in the manuscript and English language improved a lot. All the methodological questions were answered and the Ethical Approval informations was provided. Said that, I recommend the paper for publication.

Author Response

The authors thank for the approval.